# The role of physical activity in modulating six-minute walk distance in adolescents

**Attilio Carraro**[1]*, **Roberto Roklicer**[1], **Giampaolo Santi**[1], **Alessandra Colangelo**[2],
**Marco Petrini**[3], **Marta Duina**[1], **Markus Gerber**[4], **Antonino Mulè**[1]

**1** Faculty of Education, Free University of Bozen-Bolzano, Brixen-Bressanone, Italy, **2** Department FISPPA, University of Padua, Padua, Italy, **3** Marche Regional School Office, Ancona, Italy, **4** Department of Sport, Exercise and Health, University of Basel, Basel, Switzerland

* attilio.carraro@unibz.it

## Abstract

### Purpose

The six-minute walk test (6MWT) is one of the most widely utilized submaximal tests to assess cardiorespiratory fitness in both healthy and clinical populations. The present study aims to provide 6MWT reference values for Italian adolescents (11–18 years) and to identify key variables influencing the six-minute walk distance (6MWD).

### Methods

Data were collected from 3276 Italian adolescents (48% males). Participants provided self-reported age, height, weight, and weekly practice of moderate-to-vigorous physical activity (MVPA). 6MWT was performed to assess participants' cardiorespiratory fitness.

### Results

Average distances of 730.0 (IQR: 674.0–788.0) and 675.0 (IQR: 624.0–720.0) meters were walked in the 6MWT by male and female adolescents, respectively. Multiple linear regression models, used to investigate whether demographics, anthropometrics, and MVPA predicted the 6MWD, revealed statistically significant results in both males ($p < .001$, F = 28.94, adjusted $R^2 = 0.11$) and females ($p < .001$, F = 23.00, adjusted $R^2 = 0.09$). In male adolescents, MVPA and age positively predicted the 6MWD, whereas BMI negatively predicted the 6MWD. In female adolescents, MVPA positively predicted 6MWD, whereas age and BMI negatively predicted 6MWD. No interaction effects were found.

### Conclusion

The reference percentiles of distance walked, classified by sex and age, reported in the present study can be used as a practical tool to assess adolescents'

**Data availability statement:** The datasets generated for this study are available on https://doi.org/10.34740/kaggle/dsv/13016985.

**Funding:** This study was funded by the Free University of Bozen-Bolzano, grant No. 6118, as a part of the research project "PE4MOVE - L'educazione Fisica per la promozione dell'attività motoria e sportiva degli studenti nel tempo extrascolastico" [Physical Education to Promote Out-of-School Physical Activity]. The paper publication was also supported by the Open Access Publishing Fund of the Free University of Bozen-Bolzano.

**Competing interests:** The authors have declared that no competing interests exist.

cardiorespiratory fitness. This study also highlights the importance of PA engagement in this population, as it was considerably associated with the average distance walked. Consequently, in addition to anthropometric and demographic variables, future studies should pay particular attention to the amount of MVPA practiced.

## Introduction

Cardiorespiratory fitness (CRF), in a broader sense, describes a person's ability to withstand prolonged exercise. It is considered an essential and objective parameter of an individual's health, and low levels of CRF are recognized as the fourth leading risk factor for cardiovascular disease [1,2]. Higher levels of CRF have been shown to be associated with a better cardiovascular profile in both children and adolescents [3]. Over the past few decades, numerous studies have evaluated CRF in both healthy populations [4,5] and those in healthcare settings [6]. Amongst other methods, CRF is measured via the assessment of maximum oxygen uptake (VO2max) using ergo-spirometry. Direct evaluation of the $VO_2$max requires suitable facilities, specialized equipment such as treadmills or cycle ergometers, and trained laboratory staff to ensure correct test execution [7]. Consequently, conducting these tests on large population samples is challenging. To overcome this, several methods were designed for the indirect estimation of the VO2max [8–11]. One of the most widely utilized, cost-effective, and straightforward methods for the indirect estimation of the $VO_2$max is the 6-Minute Walk Test (6MWT) [12].

The 6MWT is a submaximal test widely used to assess functional exercise capacity in both healthy and clinical populations [9]. It measures the distance a subject covers walking for 6 minutes (6MWD), typically along a pathway delimited by two cones. Developed in the 1970s, the American Thoracic Society (ATS) published general guidelines in 2002 for its proper implementation in clinical and research settings. For instance, the ATS recommends a walkable corridor of 30 meters and a test-sensibilization trial to be conducted prior to the actual test [13]. Over the past decade, the test has also been widely used among children and adolescents [14–16]

The 6MWT is simple to carry out and easy to perform, as it does not require specific equipment or excessive time for administration. Additionally, being a submaximal test, it closely reflects activities of daily living, making it useful for assessing exercise tolerance in both untrained healthy individuals and those with various health issues [15–18]. According to the literature, 6MWT is also used as a prognostic tool for cardiopulmonary disorders and for monitoring the progression of different metabolic, musculoskeletal, and neuromuscular diseases [15,17,18], as well as pulmonary and cardiovascular diseases such as asthma [19] and heart failure [20]. Besides, the 6MWT has been extensively used to monitor and assess CRF in individuals with obesity [21–23].

Several studies have investigated the variables that may influence the distance covered in the test. Age, weight, height, and leg length appear to be the most influential factors [24,25]. A 2016 review confirmed that, in formulating equations for predicting the

6MWD, variables with the highest predictive significance were age, height, weight, and heart rate. In general, it is reported that older children and adolescents tend to perform better than younger ones, while a higher weight corresponds to a lower exercise capacity [15]. In a study by Saraff et al. [17], focusing on the relationship between sex, age, height and the 6MWD, it was found that older and taller children covered longer distances, but a slight decline was observed in teenage females, possibly attributable to estrogen-related increments in fat mass [17]. Conversely, Klepper and Muir [24], in their investigation of reference values for the 6MWT in children living in the United States, noted that age does not seem to be a particularly significant component, nor does the difference between males and females. Instead, factors such as motivation, exercise attitude, and the aerobic capacity of children were found to influence the distance covered. Additionally, their study showed that variations in testing procedures and the track length contribute to variations in results.

Several other confounding variables may also influence the distance covered during the 6MWT, including geographical location, environment, lifestyle, ethnicity, attitude, and motivation for the test. Consequently, researchers strongly recommend establishing specific reference values for various population groups based on different geographic or demographic contexts [24,25]. This approach can also be useful to establish a database for comparing the values obtained from examined patients, thus understanding their condition and performance capacity. In their review, Mylius et al. [16] compared data from multiple countries and found that reference values and predictive equations for the 6MWD in children and adolescents developed in one country may not be applicable or significant in others. This observation is corroborated by another review study [15], indicating a potential difference of 159 meters between values obtained in different countries. Moreover, better results were reported in middle-income countries, such as Thailand ($677 \pm 67$ m), while worse outcomes were observed in high-income countries, such as the United States ($518 \pm 73$ m), although no specific socioeconomic survey was conducted among participants, and the corridor length used for the test varied moderately, these factors may have influenced the results.

Recent studies have been conducted to provide reference values for children, adolescents, and adults in different countries. For instance, in India [26], reference values and predictive equations have been established for children, adolescents, and adults, to help prescribe exercise for patients with post-Covid dysfunctions. The importance of the 6MWT for assessing exercise tolerance in children and adolescents is constantly growing. Recent studies have confirmed its use in young people [27], although obtaining stable reference values for younger age groups is challenging due to the influential role of developmental factors and age during this life stage. Continually updating this data is crucial because the population evolves over time, along with its characteristics [15–17]. In Italy, to the best of our knowledge, only one study was conducted in 2018, providing reference values of 6MWD in healthy children aged between 6 and 11 years [18]. The study showed a strong relationship between test results, age, and anthropometric variables in children. In particular, age affects the distance covered, but weight and height must also be considered as very important factors in this growth period. In this regard, it was found that younger children (6–8 years) showed a higher correlation between 6MWD and age and anthropometric data than older children. In addition, a greater distance walked was observed in males than in females. However, no data are reported in the literature for Italian adolescents. Providing reference values is fundamental for several reasons: they help determine functional capacity levels by offering age-specific normative criteria, support the monitoring of growth and development through adolescence, and can help in the early recognition and prevention of potential future health issues.

Given the background presented above, this study aimed at (i) providing reference values for the 6MWT for adolescents aged 11–18 years, helping to bridge the literature gap in Italy, and (ii) identifying key variables that influence the 6MWD.

## Materials and methods

### Statistical analysis

Data are reported as median and interquartile range (IQR) for continuous variables (age, height, weight, BMI, and 6MWD) and as frequency (%) for categorical variables (amount of leisure-time MVPA). The boxplots method was used to detect

outlier data of 6MWD. Outliers were defined as data points falling outside the extreme fences, calculated as Q1 − 3*IQR and Q3 + 3*IQR, respectively [31]. The data normality distribution was assessed through the Shapiro-Wilk test, which revealed a non-normal distribution. Mann-Whitney test was used to compare 6MWD between males and females. Spearman correlation analyses were used to investigate possible correlations between the variables investigated in both sexes. Test result percentile curves according to age by sex stratification were reported, considering the 3$^{rd}$, 10$^{th}$, 25$^{th}$, 50$^{th}$, 75$^{th}$, 90$^{th}$, and 97$^{th}$ percentiles as proposed in the Italian study by Vandoni et al. [18]. Multiple linear regression assumptions were checked. The normality of residuals was checked by plotting the standardized residuals and the standardized predicted values, showing a normal distribution of the residuals. The multicollinearity was checked by calculating the Variance Inflation Factor (VIF) score, which was < 3 for all the independent variables. Moreover, the homoscedasticity was checked by performing the Breusch–Pagan/Cook–Weisberg test for heteroskedasticity, which showed a violation of this regression assumption only in males (p = .0003). Accordingly, a robust standard errors method was used [32]. Multiple regression analysis was performed to investigate the relationship between 6MWD (dependent variable) and age, BMI, and MVPA (independent variables). Linear, squared, and cubic multiple regressions were compared, in the two sexes separately, considering the Akaike Information Criterion (AIC) and Bayesian Information Criterion (BIC) methods, as well as the adjusted $R^2$ [33,34]. The best-fitting models in the male group were linear and cubic, while the linear model fitted best in the female group. To present the same model results for both groups, only the multiple linear regression results were reported. Interaction effects were also investigated. Statistical analyses were conducted using IBM Statistical Package for the Social Sciences – SPSS Statistics (version 29; IBM Corp., Armonk, NY: IBM Corp) and StataCorp. 2021 (release 17), the significance was set at $p = 0.05$, and the confidence intervals (CI) at 95%.

## Design and setting

The sample of the present study is a part of the PE4MOVE project, a multicentre randomised controlled trial (registration code: ISRCTN16155799) [28]. Ethical approval was obtained from the Ethic Committee of the Free University of Bozen-Bolzano (ethical approval code: P4Move Cod. 2021_01). Written consent was obtained from all participants' parents/guardians. Physical education teachers were instructed on how to perform the test with their students (e.g., after appropriate warm-up, with the right conditions) through an internet-supported continuous professional development training course [28]. The data used in this study is the pre-trial data recorded in the PE4MOVE project. As reported in the study protocol [28], pre-trial data was collected from all participants at the start of the project (i.e., at the beginning of the school year; from 27 September to 26 November 2021).

## Participants

A total of 4013 adolescents attending the lower and upper secondary school students of the Marche Region (central part of Italy) were recruited, 3276 participants (48% males) provided full personal information (age, weight and height) and took part in the test.

## Measures

**Cardiorespiratory fitness.** The 6MWT was adopted as a measure of CRF. This test has been widely used among different age groups, demonstrating to be safe and easy to perform in both children and adolescents [29]. The test was performed according to the American Thoracic Society (ATS) guidelines [13]. Participants were instructed to walk as fast as possible without running or jogging and were allowed to stop whenever they wanted. As mentioned in the ATS guidelines, participants received verbal encouragement using standardized phrases. The test was conducted in the schools' gyms. Each participant walked continuously for 6 minutes at a self-selected pace along a 20-m measured tape line, with cones placed at each end of the course. Evaluators explained the test procedures before the start. To ensure that the participants understood

the instructions, one practice trial over one track length was completed. The tests were administered by the same teachers who had undergone training dedicated to the standardization of the test procedures [28].

**Anthropometric and demographic characteristics.** Participants in the study were asked to self-report their age, sex, weight and height, and body mass index (BMI kg/m$^2$) was then calculated. According to the PE4MOVE protocol [28], it was preferred to record participants' weight and height through self-report questions, to avoid any emotional impact of objective measurements on participants from vulnerable groups.

**Physical activity levels.** Activity levels were measured using the Health Behaviour in School-aged Children (HBSC) single-item self-report measure, recognized by the WHO [30]. This instrument has been proven reliable and valid for assessing the level of moderate-to-vigorous physical activity (MVPA) practiced by school students in the last week during their leisure time. The item was the question "How many hours do you usually exercise in your free time, so much that you get out of breath or sweat?" and possible responses were: "None", "About half an hour per week", "About one hour per week", About 2-3 hours per week", "About 4-6 hours per week", and "About 7 hours per week".

## Results

The total sample consisted of 3276 adolescents, with a BMI of 19.7 (IQR: 17.8–21.8) kg/m$^2$, aged 14.0 (IQR: 12.0–16.0) years. The most frequently reported level of physical activity was 2–3 hours per week (26.7%). Eighty-three participants (2.5%, 1.4% male, 2.1% female) were excluded from the analyses as they were identified as outliers in terms of 6MWD. Therefore, 3193 adolescents (48% male, 52% female) were considered in the subsequent analyses.

The distance covered in the 6MWT ranged from 715.5 (IQR: 667.3–762.3) m for 11-year-old to 787.0 (IQR: 681.0–890.5) m for 18-year-old males and from 683.0 (644.0–730.0) m for 11-year-old to 660.0 (IQR: 597.0–753.5) m for 18-year-old females.

The 6MWD values were significantly different (p < 0.001) between male (730.00, IQR: 674.00–788.00 m) and female adolescents (675.00, IQR: 624.00–720.00 m). Spearman correlation analysis showed that in males, 6MWD was significantly correlated with age (r = .06, $p$ = .015), height (r = .08, $p$ = .001), BMI (r = −.16, $p$ < .001), and MVPA (r = .23, $p$ < .001). In females, 6MWD was significantly correlated with age (r = −.20, $p$ < .001), BMI (r = −.21, $p$ < .001), and MVPA (r = .38, $p$ < .001). Height and 6MWD were not correlated in females (r = −.03, $p$ = .221).

In males, from 11 to 18 years, the 50th percentile values resulted in 715.5 m, 728.0 m, 748.0 m, 744.0 m, 720.0 m, 700.0 m, 760.0 m, and 787.0 m, respectively. Females' percentile distribution of 6MWD at different ages. In females, from 11 to 18 years, the 50th percentile values resulted in 683.0 m, 691.0 m, 690.0 m, 680.0 m, 648.0 m, 640.0 m, 676.0 m, and 660.0 m, respectively.

### Subjects' demographic and anthropometric characteristics, and MVPA participation

For males, BMI ranged between the 50th and 75th percentile in all age categories, as well as for females aged 11–17 years. In contrast, 18-year-old females showed a BMI slightly below the 50th percentile (according to cut-off value proposed by the National Center for Health Statistics in collaboration with the National Center for Chronic Diseases Prevention and Health Promotion). Since the BMI was between the 5th and the 85th percentile, the sample can be considered healthy weight. Table 1a shows males' anthropometric and demographic data, and 6MWD values for all ages.

Table 1b shows females' anthropometric and demographic characteristics, and 6MWD values for all ages.

### Percentile curves according to age by sex stratification

Fig 1a and 1b show the percentiles range from 3rd to 97th, of 6MWD at different ages in males (1a) and females (1b).

### Linear regression analysis

In both males and females, the linear regression analyses revealed statistically significant results ($p$ < .001, F = 28.94, adjusted R$^2$ = 0.11 and $p$ < .001, F = 23.00, adjusted R$^2$ = 0.09; males and females respectively). In males (Table 2), leisure

**Table 1. a. Anthropometric, demographic, and 6MWD results in males stratified by age. b. Anthropometric, demographic, and 6MWD results in females stratified by age.**

| Age | n | MVPA | Weight | Height | BMI | 6MWD |
|---|---|---|---|---|---|---|
| a | | | | | | |
| 11 | 220 | 7.3% | 41.0 (35.3–48.0) | 149.5 (143.3–154.0) | 18.5 (6.7–20.4) | 715.5 (667.3–762.3) |
| 12 | 387 | 17.8% | 45.0 (40.0–52.0) | 155.0 (150.0–162.0) | 18.7 (17.0–20.8) | 728.0 (675.0–799.0) |
| 13 | 182 | 18.1% | 54.0 (47.0–60.3) | 164.5 (158.0–170.0) | 19.9 (17.9–21.8) | 748.0 (690.0–815.8) |
| 14 | 185 | 29.7% | 58.0 (52.0–65.0) | 170.0 (164.0–175.0) | 20.1 (18.3–22.0) | 744.0 (680.0–812.5) |
| 15 | 191 | 18.8% | 61.0 (55.0–69.0) | 175.0 (170.0–179.0) | 20.2 (18.3–22.2) | 720.0 (648.0–780.0) |
| 16 | 166 | 28.9% | 65.0 (59.0–74.0) | 176.0 (171.8–180.0) | 21.1 (19.0–22.9) | 700.0 (640.0–748.8) |
| 17 | 128 | 26.6% | 65.0 (62.0–77.0) | 178.0 (174.0–181.0) | 21.2 (19.6–23.9) | 760.0 (680.0–828.8) |
| 18 | 76 | 31.6% | 70.0 (63.0–78.8) | 180.0 (174.0–184.0) | 21.9 (19.9–23.3) | 787.0 (681.0–890.5) |
| Total | 1535 | 20.5% | 55.0 (45.0–65.0) | 166.0 (155.0–175.0) | 19.7 (17.8–22.0) | 730.0 (674.0–788.0) |
| b | | | | | | |
| 11 | 236 | 5.5% | 40.0 (35.0–46.0) | 150.0 (145.0–155.0) | 17.8 (16.0–20.4) | 683.0 (644.0–730.0) |
| 12 | 369 | 8.9% | 45.0 (40.0–52.0) | 156.0 (150.0–162.0) | 18.4 (16.9–20.6) | 691.0 (648.0–743.0) |
| 13 | 176 | 15.9% | 50.0 (45.0–56.0) | 160.0 (156.0–165.0) | 19.5 (17.9–21.7) | 690.0 (640.8–752.0) |
| 14 | 161 | 10.6% | 53.0 (48.0–59.0) | 163.0 (158.0–167.0) | 20.1 (18.5–21.8) | 680.0 (620.0–720.0) |
| 15 | 259 | 11.6% | 54.0 (49.0–60.0) | 164.0 (159.0–168.0) | 20.3 (18.4–22.5) | 648.0 (600.0–700.0) |
| 16 | 247 | 8.9% | 55.0 (51.0–60.0) | 164.0 (160.0–168.0) | 20.6 (19.1–22.5) | 640.0 (600.0–686.0) |
| 17 | 161 | 11.8% | 56.0 (50.0–63.0) | 164.0 (160.0–168.0) | 20.8 (18.8–22.8) | 676.0 (601.5–720.0) |
| 18 | 49 | 0% | 55.0 (51.0–61.0) | 163.0 (160.0–167.5) | 20.6 (19.1–23.0) | 660.0 (597.0–753.5) |
| Total | 1658 | 9.8% | 50.0 (45.0–57.0) | 160.0 (155.0–165.0) | 19.7 (17.7–21.6) | 675.0 (624.0–720.0) |

Data are reported as median (IQR) and percentage. Age is reported in years, weight in kg, height in cm, BMI in kg/m², and 6MWD in meters. BMI = Body Mass Index, MVPA = percentage of participants who performed at least 7 h/week of leisure time moderate-vigorous physical activity - MVPA.

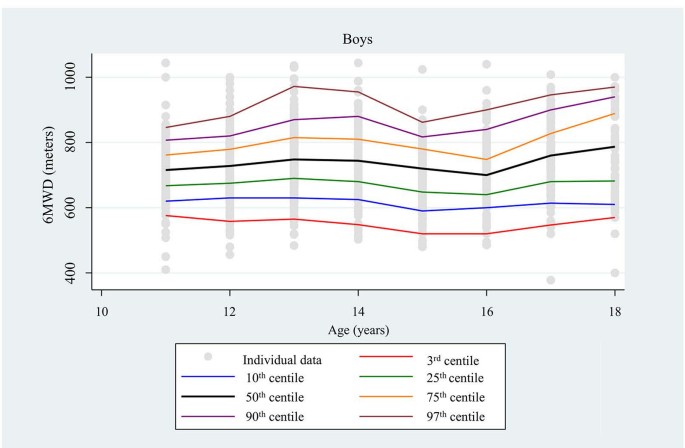
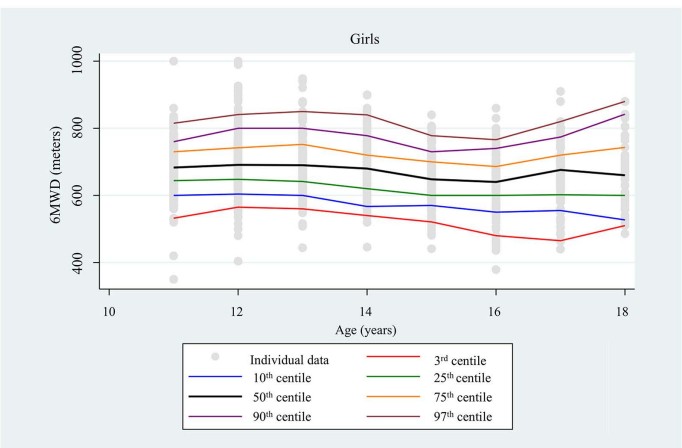

**Fig 1. Males' percentile distribution of 6MWD at different ages.**

**Table 2. Linear regression analyses in both sexes.**

| Group | Outcome variable | B | t | p | 95% CI |
|---|---|---|---|---|---|
| Males | 1 h/week MVPA | 23.197 | 2.31 | .021 | 3.519 to 42.874 |
| | 2-3 h/week MVPA | 23.963 | 2.67 | .008 | 6.370 to 41.502 |
| | 4-6 h/week MVPA | 44.682 | 5.01 | <.001 | 27.199 to 62.164 |
| | ≥ 7 h/week MVPA | 68.884 | 7.37 | <.001 | 50.548 to 87.220 |
| | Age | 6.417 | 5.04 | <.001 | 3.921 to 8.914 |
| | BMI | −6.528 | −9.23 | <.001 | −7.916 to −5.141 |
| Females | 2-3 h/week MVPA | 24.366 | 4.18 | <.001 | 12.920 to 35.813 |
| | 4-6 h/week MVPA | 39.281 | 6.16 | <.001 | 26.781 to 51.782 |
| | ≥ 7 h/week MVPA | 46.891 | 6.17 | <.001 | 31.989 to 61.792 |
| | Age | −5.580 | −5.78 | <.001 | −7.472 to −3.688 |
| | BMI | −3.558 | −5.46 | <.001 | −4.836 to −2.279 |

BMI = Body Mass Index, MVPA = leisure time moderate-to-vigorous physical activity.

time MVPA and age positively predicted the 6MWD, while BMI negatively predicted the 6MWD. In female groups (Table 2), leisure time MVPA positively predicted the 6MWD. In contrast, age and BMI negatively predicted the 6MWD. No interaction effects were found in both male and female models.

## Discussion

Although several studies have provided reference values for 6MWD in children [15,17,18,24,26,35] and adolescents [15,20,26,27,35–37], to the best of our knowledge, the present study is the first to provide reference values of the 6MWT for Italian adolescents aged 11–18 years.

Based on our sample, which consisted of Italian adolescents within a normal BMI range, only a small proportion met the WHO recommendation for MVPA. Specifically, 15% of the total sample achieved at least 7 hours per week of leisure-time MVPA, with 20.6% of males and 9.7% of females meeting this threshold. Sex differences in youth are commonly observed, particularly in cardiorespiratory fitness [4,38]. Several biological mechanisms may contribute to these differences, including variations in muscle fiber type, oxygen extraction efficiency, and myofibrillar lipid content [39–41]. Considering these sex-related physiological differences may help contextualize observed disparities in physical activity and fitness levels between male and female adolescents. Furthermore, the low percentage of people who met the WHO recommendations for MVPA may be related to several factors. Evidence showed the absence of immediate physical benefits may lead to a reduction in motivation for leisure-time physical activity in this population. Urban environments often lack safe, accessible outdoor spaces, while smartphones, streaming, and video games foster sedentary lifestyles. Moreover, academic pressures along with digital and social priorities further deprioritize exercise [42]. While these factors likely contribute, they were not directly assessed in this study.

Consistent with our findings, a study conducted on Slovenian adolescents reported a decline in overall PA, with a more pronounced reduction in MVPA observed among females [43]. The present study showed a median 6MWD of 730 meters for males and 675 meters for females. These values are similar to those reported in a previous review [16]. Specifically, in females, 6MWDs in the present study were comparable with results observed among adolescents from Switzerland (635 m), Sweden (650 m), Austria (663 m), Tunisia (725 m) and Brazil (705 m). However, the results of our sample were slightly higher than those found among adolescents from Taiwan (503 m) and Turkey (538 m). Conversely, for males, 6MWD results obtained in the present study were somewhat higher compared to those reported for adolescents living in Taiwan

(561 m), and Turkey (556 m), while similar to those found in Switzerland (665 m), Tunisia (758 m), Austria (698 m), and Brazil (711 m).

Among adolescent samples, few studies have focused on the relationship between 6MWD and anthropometric/demographic variables, reporting contrasting results. Kanburoglu et al. [36] investigated this relationship in adolescents aged 12–18 years, showing an inverse correlation between 6MWD and height, weight, and BMI, in both sexes. However, regression analysis showed no significant influence of these variables on the test results. Although it is not possible to distinguish between children or adolescents as the authors performed an age-adjusted correlation analysis, Geiger et al. [29] found that 6MWD correlated positively with height in both sexes and negatively with BMI in males. Rahman et al. [37] also studied this relationship in adolescents aged 12–18 years, reporting a positive correlation between 6MWT results and age, height, and weight. In addition, Ulrich et al. [35] found that age was the only significant predictor of 6MWD in adolescent males (13–16 years old), whereas in females (12–16 years old), both age and weight were significant predictors. Our findings are partially consistent with this existing body of evidence. Specifically, age, height, and BMI were significantly correlated with 6MWD in males, whereas only age and BMI were correlated with 6MWD in females. The different results in the studies may be related to the physical fitness levels of the participants. So far, few studies have considered participants' PA habits. Kanburoglu et al. [36] found that participants' PA levels were significantly associated with 6MWD. They stratified the sample according to the PA levels of participants (very active: ≥ 30 min, 5 or more times/week; active: ≥ 30 min, 3 or more times/week; sedentary: < 30 min, or 3 times/week) and showed that sedentary male students had lower 6MWD results than very active students, whereas no significant differences were found between active and very active students across both sexes. Ulrich et al. [35] found that the physical activity score (PAS) was an important predictor of 6MWD in adolescent females. However, analyses stratified by sex revealed that PAS correlated with 6MWD only in male children and adolescents. Our results are in agreement with those reported in the studies described above. Specifically, the model showed that the practice of 1–7 or more hours/week of leisure time MVPA positively predicted 6MWD in males. In females, involvement in 2–7 or more hours/week of leisure-time MVPA and height positively predicted 6MWD. In addition, some other factors influencing physical test results may be associated with the specific life stage, as adolescents are often facing a lack of motivation in physical activity during their leisure time [44]. It was surprising to observe a decline in 6MWT results among participants aged 13–16 years. Another study indicated an evident decline in PA levels throughout adolescence [45]. This trend was also reported by Ulrich et al. [35] and Kanburoglu et al. [36], who hypothesized that it is related to hormonal changes during adolescence and different levels of intrinsic motivation among teenagers at different ages. Another explanation for this phenomenon may be the strong relationship between 6MWD and leisure-time MVPA found in the present study. This is further supported by the review of Hills et al. [46], who described that regular participation in MVPA increases CRF in young people.

Adolescence is a challenging period of life, with various factors affecting personal growth [47]. PA is one such factor and, as generally reported in the literature, has a strong relationship with individual physical growth and all physical test scores. Lack of PA is associated with a lower level of CRF [48], thereby negatively affecting cardiovascular risk factors [49,50]. Moreover, physiological and psychological health has been found to improve with regular engagement in MVPA throughout adolescence [51]. For this reason, it is important to consider PA in similar future studies. Further research is needed to better understand and explain the relationship between 6MWD, anthropometric, demographic characteristics, and PA participation in adolescents, as well as to elucidate the variables underlying the different results reported in the literature.

This study provides useful information on reference values of 6MWD in Italian adolescents. Strengths of the study are the large sample size, along with results reported for both sexes. Another strength of the study is the consideration of participation in MVPA during the week, as it can be an extremely influential factor. Nevertheless, this study has some limitations. More specifically, the population recruited for this study was limited to residents of a single Italian region (Marche), which may compromise the generalizability of the findings. Another limitation is the use of self-reported measurements

of anthropometric and MVPA variables, although the test-retest reliability of HBSC in adolescents was good for a self-administered tool, it is lower than objective PA monitoring tools [52,30].

Finally, while the excluded data represent a minor fraction of the overall sample, their omission may have resulted in a slight selection bias, though unlikely to affect the overall findings.

## Conclusions

The present study provides 6MWT reference values for Italian adolescents aged 11–18 years. Our results indicate that age, height, and BMI positively correlated with 6MWD in males, while age and BMI negatively correlated with 6MWD in females. PA levels also predicted 6MWD, as higher activity levels were associated with better performance. Nevertheless, only a small percentage of the total sample of respondents met the MVPA recommendations proposed by the WHO. Surprisingly, we found a decline in 6MWT results among participants aged 13–16 years. This study highlights the importance of PA engagement among adolescents, as it influenced the average distance walked. Further research is needed to confirm this, as well as to deeply explore the relationship between 6MWD and various demographic, anthropometric, and PA variables among adolescents.

## Acknowledgments

The authors are grateful to the teachers and PE coordinators of the secondary schools of the Marche Region for their contributions to the implementation of the study.

## Author contributions

**Conceptualization:** Attilio Carraro.

**Formal analysis:** Attilio Carraro, Roberto Roklicer, Giampaolo Santi, Marta Duina, Antonino Mulè.

**Funding acquisition:** Attilio Carraro.

**Investigation:** Alessandra Colangelo, Marco Petrini.

**Methodology:** Attilio Carraro.

**Project administration:** Attilio Carraro.

**Supervision:** Attilio Carraro.

**Visualization:** Attilio Carraro, Antonino Mulè.

**Writing – original draft:** Attilio Carraro, Roberto Roklicer, Giampaolo Santi, Marta Duina, Antonino Mulè.

**Writing – review & editing:** Attilio Carraro, Roberto Roklicer, Giampaolo Santi, Alessandra Colangelo, Marco Petrini, Marta Duina, Markus Gerber, Antonino Mulè.

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
