## [Decision Letter · Decision Letter 0]

14 Aug 2025

Dear Dr. Carraro,

Thank you for submitting your manuscript to PLOS ONE. After careful consideration, we feel that it has merit but does not fully meet PLOS ONE’s publication criteria as it currently stands. Therefore, we invite you to submit a revised version of the manuscript that addresses the points raised during the review process.

Please carefully consider the detailed observations provided by the reviewers

We look forward to receiving your revised manuscript.

Kind regards,

Andrea Martinuzzi

Academic Editor

PLOS ONE

“This study was funded by the Free University of Bozen-Bolzano, grant No. 6118, as a part of the research project “PE4MOVE - L’educazione Fisica per la promozione dell’attività motoria e sportiva degli studenti nel tempo extrascolastico” [Physical Education to Promote Out-of-School Physical Activity]. The paper publication was also supported by the Open Access Publishing Fund of the Free University of Bozen-Bolzano.”

4. In the online submission form you indicate that your data is not available for proprietary reasons and have provided a contact point for accessing this data. Please note that your current contact point is a co-author on this manuscript. According to our Data Policy, the contact point must not be an author on the manuscript and must be an institutional contact, ideally not an individual. Please revise your data statement to a non-author institutional point of contact, such as a data access or ethics committee, and send this to us via return email. Please also include contact information for the third party organization, and please include the full citation of where the data can be found.

5. We note that there is identifying data in Tables 1a and 1b. Due to the inclusion of these potentially identifying data, we have removed this file from your file inventory. Prior to sharing human research participant data, authors should consult with an ethics committee to ensure data are shared in accordance with participant consent and all applicable local laws.

-Location data

Please remove or anonymize all personal information, ensure that the data shared are in accordance with participant consent, and re-upload a fully anonymized data set. Please note that spreadsheet columns with personal information must be removed and not hidden as all hidden columns will appear in the published file.

Reviewers' comments:

Reviewer's Responses to Questions

**Comments to the Author**

1. Is the manuscript technically sound, and do the data support the conclusions?

Reviewer #1: Yes

Reviewer #2: Yes

2. Has the statistical analysis been performed appropriately and rigorously?

Reviewer #1: No

Reviewer #2: Yes

3. Have the authors made all data underlying the findings in their manuscript fully available?

Reviewer #1: No

Reviewer #2: Yes

4. Is the manuscript presented in an intelligible fashion and written in standard English?

Reviewer #1: Yes

Reviewer #2: Yes

Reviewer #1: Thank you for the opportunity to review the manuscript PONE-D-25-18111, “The role of physical activity in modulating six-minute walk distance in adolescents”. The article presents original research on the determinants of six-minute walk distance (6MWD) in adolescents. Several issues in clarity, terminology, methodological reporting, and contextualization must be addressed before the manuscript is suitable for publication. My specific comments are organized by manuscript section and line number below.

Introduction

1. Lines 47 & 56 - The dichotomy “healthy and clinical populations” is unclear. The term clinical is inappropriate here. Please specify which clinical conditions are being contrasted with healthy cohorts, or replace with more precise language.

2. Line 61 - The phrase “In the last years” is vague. Please specify the time period (e.g. “Between 2015 and 2022, the …”).

3. Lines 54–69 - The categorical grouping of conditions (metabolic, pulmonary, cardiovascular, musculoskeletal) is incorrect (e.g., asthma is pulmonary, not metabolic or musculoskeletal). Revise or clarify to ensure each condition is referred to or classified correctly.

4. Lines 72–74 - Change “Most considered variables” to “Variables with the greatest predictive weight” (or similar).

5. Lines 88–89 - Revise “different contexts” to “different geographic or demographic contexts,” since variation may occur within a single country.

6. Lines 88–96 - The terms “reference values,” “equations,” and “population standards” are used without sufficient contextualisation. Be more specific when referring to these factors – (e.g equations for what? population standards for what?). Explain what outcomes the equations predict, and cite recent literature justifying the need for improved 6MWD prediction models.

7. Lines 95–98 - The assertion that “better results were reported in middle-income countries” lacks substantiation. Provide references, acknowledge the absence of socioeconomic data in prior studies, and offer plausible explanations for observed differences.

8. Lines 99–117 - The rationale for conducting this study in Italy is underdeveloped in this section. Please review relevant European or Italian adolescent health literature to establish the study’s necessity and contextualize the findings. Furthermore, while the authors acknowledge the link between cardiovascular health and 6MWD – there is little rationale explaining why these two are relevant to the population understudy.

Methods

Measures

• Line 139 - Define “CRF” (cardiorespiratory fitness) at first mention.

Statistical Analyses

1. Line 166 - Excluding participants without complete personal data may introduce selection bias. Acknowledge this in the limitations.

2. Line 169 - A non-normal distribution was noted, yet results are summarized using means and standard deviations. Report medians and interquartile ranges instead, or justify why mean ± SD are appropriate.

3. Lines 169–180 - The use of a linear regression analysis appears unsubstantiated and requires elaboration. Given the non-normal distribution. the data is likely to violate several assumptions that are essential to consider when conducting a linear regression. Specify which regression assumptions were tested (e.g., normality of residuals, homoscedasticity), present these diagnostics in a supplement, and cite best practices for outlier removal and model selection criteria (AIC/BIC).

It remains unclear whether all predictors were entered into a single regression model. Including highly correlated variables (height, weight, and BMI) violates the independence assumption, introduces multicollinearity, inflates standard errors, and undermines interpretability. If BMI is included alongside height and weight, the model essentially attempts to partition shared variance among highly correlated predictors, violating the assumption of independence among predictors and potentially leading to misleading conclusions. I strongly recommend revisiting the model either (a) modelling BMI alone or (b) modelling height and weight separately with age to determine interactions.

Results

1. Line 188 - Clarify whether the reported BMI (“20.00 ± 3.20 kg/m²”) is mean ± SD or median (IQR). Could the authors include more information on BMI and how it reflects overall health status in this context? As it stands, it is difficult to assess whether the individuals in the study can be considered healthy, especially given the lack of clinical or physiological markers beyond BMI. Clarifying this would help readers better understand the health profile of the study population – and provide context to the utility of MVPA as a marker of CRF.

2. Line 190 - The statement regarding WHO-recommended MVPA levels belongs in the Discussion.

3. Line 190 - Justify the exclusion of outliers by age group and provide details in a supplement.

4. Line 201 - When reporting sex differences in 6MWD, specify whether these differences apply across all ages or vary by age subgroup.

5. Line 214 - Clarify whether “6MWT” differs from “6MWD” or if the terms are used interchangeably.

6. Figure 1a & 1b - Add individual data points or provide supplementary scatterplots to allow readers to assess distribution and potential clustering.

Discussion

1. Lines 223–240 – The authors need to re-evaluate their models, and ensure their discussion addresses possible interaction effects.

2. Lines 250–251 - When comparing your results to those from Acuri et al. (2016) and other countries, discuss potential cultural, socioeconomic, or environmental factors that may account for observed similarities or differences between Italian adolescents and peers in Switzerland, Tunisia, Austria, and Brazil.

3. Line 273 - The phrase “state of physical fitness” is vague. Specify whether you refer to objectively measured fitness, self-reported activity levels, or other proxies, and discuss potential biases or population differences.

4. Line 274 & 283–285 - Define “PA” (physical activity) when first used. Consider interactions between MVPA and BMI in your models - does BMI moderate the effect of MVPA on 6MWD? Also discuss how gendered social norms, environmental opportunities, and motivational factors might influence activity levels.

5. Interpretation of Correlations - The positive correlations of age, height, and BMI with 6MWD in males (versus negative in females) warrant deeper biological, social, and cultural contextualization in light of adolescent growth trajectories and gender norms relating to health and fitness.

6. MPVA Findings - Discuss why MVPA levels fell below WHO recommendations despite the study population being a seemingly healthy cohort.

7. Adolescent 6MWD Decline - Explore literature addressing the decline in 6MWD between ages 13–16, linking to pubertal development, motivational factors, or methodological considerations.

Limitations

• Several limitations are not addressed, including selection bias from data exclusions, potential measurement error in MVPA assessment, and the generalizability of findings beyond a high-income, European context.

• Furthermore, while the authors mention some limitations they do not elaborate on how these may have influenced the study’s design or findings.

Conclusion

The manuscript addresses a valuable research question, but substantial revisions are needed to clarify terminology, strengthen methodological transparency, ensure statistical validity, and properly contextualize. I would be happy to review a revised version of the manuscript that addresses these concerns.

Reviewer #2: Authors covered a gap in the literature related to the normative data in Italian adolescents about 6MWT. The sample is big and representative of the agegroup. However it has been colleceted only in one region as mentioned as limitation by the authors. I have some doubt about the self report measure. Authgors should spend some more words in the discussion about the limitation of the study, since it has been reported in the cited paper (Booth; the physical activity questions in the WHO health behaviour in schoolchildren (HBSC) survey) that there is an agrement from 70% to 85% in test retest. Moreover some references about the reliability with objective measure should be reported. Also self reporting heigth and weigth could introduce biases. In table 2 it is not reported the regression for age in males. Please add.

**Do you want your identity to be public for this peer review?** For information about this choice, including consent withdrawal, please see our Privacy Policy

Reviewer #1: **Yes:** Elizabeth S. Dinkele

Reviewer #2: No

---

## [Author Response · Author response to Decision Letter 1]

24 Sep 2025

REVIEWER 1

We would like to thank the reviewer for their constructive feedback.

Introduction

• Lines 47 & 56 - The dichotomy “healthy and clinical populations” is unclear. The term clinical is inappropriate here. Please specify which clinical conditions are being contrasted with healthy cohorts, or replace with more precise language.

Authors reply: Thank you for your comment. The sentence has been slightly rephrased to use the term “healthcare settings”, in line with the cited article.

• Line 61 - The phrase “In the last years” is vague. Please specify the time period (e.g. “Between 2015 and 2022, the …”).

Authors reply: Thank you for your comment. The sentence has been rephrased to more clearly delineate the temporal scope of the test's application among specific populations.

• Lines 54–69 - The categorical grouping of conditions (metabolic, pulmonary, cardiovascular, musculoskeletal) is incorrect (e.g., asthma is pulmonary, not metabolic or musculoskeletal). Revise or clarify to ensure each condition is referred to or classified correctly.

Authors reply: Thank you for this useful comment. The part of the sentence referring to these conditions has been revised to enhance readability and improve the overall clarity related to these conditions.

• Lines 72–74 - Change “Most considered variables” to “Variables with the greatest predictive weight” (or similar).

Authors reply: Thank you for your comment. This part is now changed, and it reads “variables with the highest predictive significance”.

• Lines 88–89 - Revise “different contexts” to “different geographic or demographic contexts,” since variation may occur within a single country.

Authors reply: Thank you for the comment. This sentence is now revised, and it reads as follows: “Consequently, researchers strongly recommend establishing specific reference values for various population groups based on different geographic or demographic contexts”.

• Lines 88–96 - The terms “reference values,” “equations,” and “population standards” are used without sufficient contextualisation. Be more specific when referring to these factors – (e.g equations for what? population standards for what?). Explain what outcomes the equations predict, and cite recent literature justifying the need for improved 6MWD prediction models.

Authors reply: Thank you for this useful comment. The sentence has been rephrased, and it clearly states what each of these terms refers to. The sentence now reads as follows: “In their review, Mylius et al. [16] compared data from multiple countries and found that reference values and predictive equations for the 6MWD in children and adolescents developed in one country may not be applicable or significant in others”.

• Lines 95–98 - The assertion that “better results were reported in middle-income countries” lacks substantiation. Provide references, acknowledge the absence of socioeconomic data in prior studies, and offer plausible explanations for observed differences.

Authors reply: Thank you for your comments. The sentence has now been clarified by incorporating the respective values recorded in each country. Additionally, the potential reasons underlying the observed differences between the two countries have been addressed.

• Lines 99–117 - The rationale for conducting this study in Italy is underdeveloped in this section. Please review relevant European or Italian adolescent health literature to establish the study’s necessity and contextualize the findings. Furthermore, while the authors acknowledge the link between cardiovascular health and 6MWD – there is little rationale explaining why these two are relevant to the population understudy.

Authors reply: Thank you for your comment. By measuring the 6MWD, this submaximal exercise test assesses aerobic capacity and endurance without pushing individuals to their absolute limit, thus making it ideal for adolescents. Although it is not a direct measure, 6MWD has shown a strong correlation with maximal oxygen consumption, which is considered a gold standard for cardiorespiratory fitness and, indeed, a relevant measure of cardiovascular health. Furthermore, in this paragraph, we also intended to reflect on research done on a sample of younger participants (PMCID: PMC6188863). Given the absence of reported data on Italian adolescents in the existing literature, a significant gap remains in research addressing this population. Building on this foundation, we infer that there is a compelling rationale for extending the investigation of these parameters to older populations. An explanation of the importance of providing reference values was also added.

Methods

Measures

• Line 139 - Define “CRF” (cardiorespiratory fitness) at first mention.

Authors reply: Thank you for your suggestion; however, the CRF is defined in the introduction part. Hence, we agreed that there is no need to define it also in the Measures.

Statistical Analyses

• Line 166 - Excluding participants without complete personal data may introduce selection bias. Acknowledge this in the limitations.

Authors reply: Thank you for your comment. Full personal data referred to participants’ age, weight, and height. Without these data, it was not possible to perform the main analyses. We apologise for not being completely clear about that. The sentence has been revised.

• Line 169 - A non-normal distribution was noted, yet results are summarized using means and standard deviations. Report medians and interquartile ranges instead, or justify why mean ± SD are appropriate.

Authors reply: Thank you for your comment. The results were presented as mean ± standard deviation since the mean and median showed similar values. The difference between median and mean was 0.19, -1.23, -0.09, -0.32, and -1.56 for age, weight, height, BMI and 6MWD, respectively. So, the mean values were not affected by outliers. Data are now presented according to the reviewer’s suggestion.

• Lines 169–180 - The use of a linear regression analysis appears unsubstantiated and requires elaboration. Given the non-normal distribution. the data is likely to violate several assumptions that are essential to consider when conducting a linear regression. Specify which regression assumptions were tested (e.g., normality of residuals, homoscedasticity), present these diagnostics in a supplement, and cite best practices for outlier removal and model selection criteria (AIC/BIC). It remains unclear whether all predictors were entered into a single regression model. Including highly correlated variables (height, weight, and BMI) violates the independence assumption, introduces multicollinearity, inflates standard errors, and undermines interpretability. If BMI is included alongside height and weight, the model essentially attempts to partition shared variance among highly correlated predictors, violating the assumption of independence among predictors and potentially leading to misleading conclusions. I strongly recommend revisiting the model either (a) modelling BMI alone or (b) modelling height and weight separately with age to determine interactions

Authors reply: Thank you for your valuable feedback. It was a mistake. The regression assumptions were checked as follows. The normality of residuals was also checked by plotting the standardized residuals and the Standardized Predicted Values. As shown in the following figures, a good normality of residuals was found.

The multicollinearity was checked by calculating the Variance Inflation Factor (VIF) score, which was 1.7, 1.7, 1.1 and 1.0 for age, height, BMI and MVPA, respectively. Moreover, to better investigate possible multicollinearity between two highly related variables (BMI and height), Spearman's correlation was performed, showing a low positive Spearman's correlation (.344). Based on these results, the multicollinearity assumption was satisfied. The homoscedasticity was checked by performing the Breusch–Pagan/Cook–Weisberg test for heteroskedasticity. The results showed a violation of this regression assumption (p=.0019). For this reason, a Robust Standard Errors method was used.

In female group, the normality of residuals was also checked by plotting the standardized residuals and the Standardized Predicted Values. As shown in the following figures, a good normality of residuals was found.

The multicollinearity was checked by calculating the Variance Inflation Factor (VIF) score, which was 1.5, 1.4, 1.1 and 1.0 for age, height, BMI and MVPA, respectively. Moreover, to better investigate possible multicollinearity between two highly related variables (BMI and height), Spearman's correlation was performed, showing a low positive correlation (.204). Based on these results, the multicollinearity assumption was satisfied.

The models, results and discussion have been modified according to reviewers’ suggestions.

Results

• Line 188 - Clarify whether the reported BMI (“20.00 ± 3.20 kg/m²”) is mean ± SD or median (IQR).

Authors reply: The data were reported as mean ± SD. It is now expressed as median (IQR).

Could the authors include more information on BMI and how it reflects overall health status in this context? As it stands, it is difficult to assess whether the individuals in the study can be considered healthy, especially given the lack of clinical or physiological markers beyond BMI. Clarifying this would help readers better understand the health profile of the study population – and provide context to the utility of MVPA as a marker of CRF.

Authors reply: Thank you for your comment. We agree with the reviewer, it is essential to collect data relating to the physical, mental, emotional and social spheres to define a child, adolescent, adult or elderly person as healthy. Unfortunately, we did not collect this data. BMI could only be used to determine if a person is underweight, healthy weight, overweight, or obese. Moreover, as reported by the National Center for Health Statistics in collaboration with the National Center for Chronic Diseases Prevention and Health Promotion, the BMI cut-off values differ according to age and sex. Consequently, we can only describe the BMI status considering these two variables. Some additional information on the participants' BMI condition is now added in the results paragraph.

• Line 190 - The statement regarding WHO-recommended MVPA levels belongs in the Discussion.

Authors reply: Thank you, we appreciate this comment. The part with the WHO recommendations for MVPA is now included in the second paragraph of the discussion. It has been slightly revised to more clearly highlight the significance of these percentages, as well as the walking distances recorded for male and female participants.

• Line 190 - Justify the exclusion of outliers by age group and provide details in a supplement.

Authors reply: As described in the statistical analysis section, boxplots method was used to detect outlier data of 6MWD and values > 3 IQR’s from the end of the box were identified as outliers and were excluded from the data analyses. The 1.5 IQR rule is a widely accepted as conventional method for outlier detection. By extending this to a 3 IQR rule, a more conservative approach was adopted (doi: 10.1080/10691898.2011.11889610). This method was chosen because it's a non-parametric approach that is less sensitive to extreme values compared to methods based on the mean and standard deviation. By applying this criterion, only the most extreme data points, which could potentially skew our results, were excluded from the final analysis, avoiding the accidental exclusion of valid data points that are merely on the tail ends of the distribution. The sentence has been modified to improve clarity.

• Line 201 - When reporting sex differences in 6MWD, specify whether these differences apply across all ages or vary by age subgroup.

Authors reply: Thank you for your comment. Speaking generally, differences between males and females in youth are very common, especially in terms of cardiorespiratory fitness (doi:10.1161/CIR.0000000000000866, doi: 10.5604/01.3001.0053.7363, doi: 10.3389/fped.2021.657551). Potential explanations for this difference include sex-related differences in muscle fibre type, oxygen extraction, or the lipid content of myofibrils (doi: 10.1016/j.jsams.2008.05.006, doi: 10.1152/ajpregu.00472.2006). To answer the reviewer's comment, we performed the analyses to investigate whether the differences between the two sexes apply across all ages. After the assumptions check, linear regression with robust standard errors was performed. Results showed a significant interaction between gender and age (B: -11.027, t: -7.45, p<0.001, CI: -13.929 – -8.126). However, since these analyses extend beyond the primary scope of the study, we have jointly decided to omit them from the manuscript.

• Line 214 - Clarify whether “6MWT” differs from “6MWD” or if the terms are used interchangeably.

Authors reply: Thank you for your comment. Although 6MWT and 6MWD are closely related, they refer to different aspects of the same assessment. 6MWT – stands for the standardized submaximal test protocol used to assess the functional capacity. 6MWD is a result of this test – total distance walked during the 6 minutes usually expressed in meters (m). To put it simple way: 6MWT is the test and 6MWD is the score.

• Figure 1a & 1b - Add individual data points or provide supplementary scatterplots to allow readers to assess distribution and potential clustering.

Authors reply: Thank you for your suggestion. The graphs have been changed.

Discussion

• Lines 223–240 – The authors need to re-evaluate their models, and ensure their discussion addresses possible interaction effects.

Authors reply: Thank you for your comment. Based on the previous suggestions, we have re-evaluated the models and the possible interaction effects are now addressed.

• Lines 250–251 - When comparing your results to those from Acuri et al. (2016) and other countries, discuss potential cultural, socioeconomic, or environmental factors that may account for observed similarities or differences between Italian adolescents and peers in Switzerland, Tunisia, Austria, and Brazil.

Authors reply: Thank you for your comment. We agree that these similarities and differences could have been influenced by cultural, socioeconomic and environmental factors. However, neither Arcuri et al. (2016) nor other authors reporting results from their respective countries have thoroughly examined these factors in their work. While these variables may hold relevance, they were not investigated as part of the present study. Therefore, we could not discuss these factors in detail.

• Line 273 - The phrase “state of physical fitness” is vague. Specify whether you refer to objectively measured fitness, self-reported activity levels, or other proxies, and discuss potential biases or population differences.

Authors reply: Thank you for your comment. This is a very good point. In this context, physical fitness refers to each participant’s individual fitness level. It represents a personal state shaped by daily lifestyle habits, rather than a directly measured variable. For instance, while some individuals may engage in regular leisure-time physical activity, others may not, reflecting natural variation in everyday behaviour.

• Line 274 & 283–285 - Define “PA” (physical activity) when first used. Consider interactions between MVPA and BMI in your models - does BMI moderate the effect of MVPA on 6MWD?

Authors reply: Thank you very much for your comment. Physical activity (PA) is now defined at its first mention in the text, ensuring clarity for the reader throughout the document. Performing the suggested models, results showed no interaction effects.

Also discuss how gendered social norms, environmental opportunities, and motivational factors might influence activity levels.

Authors reply: Thank you very much for your comment. We really appreciate the idea of discussing the gendered social norms, environmental opportunities, motivational factors and their possible effect on activity levels. However, as the aim of the study was to establish the reference values of 6MWD in adolescents, we argue that broadening the work in this direction could divert the attention of the reader from the ce

---

## [Decision Letter · Decision Letter 1]

29 Jan 2026

Dear Dr. Carraro,

We look forward to receiving your revised manuscript.

Kind regards,

Andrea Martinuzzi

Academic Editor

PLOS One

Journal Requirements:

Additional Editor Comments:

Please take care of the last 3 minor changes requested by the reviewer 1

Reviewer's Responses to Questions

**Comments to the Author**

Reviewer #1: All comments have been addressed

Reviewer #2: All comments have been addressed

2. Is the manuscript technically sound, and do the data support the conclusions?

Reviewer #1: Yes

Reviewer #2: Yes

3. Has the statistical analysis been performed appropriately and rigorously?

Reviewer #1: Yes

Reviewer #2: Yes

4. Have the authors made all data underlying the findings in their manuscript fully available?

Reviewer #1: Yes

Reviewer #2: Yes

5. Is the manuscript presented in an intelligible fashion and written in standard English?

Reviewer #1: Yes

Reviewer #2: Yes

Reviewer #1: I would like to commend the authors of this manuscript, and on their efforts to answer and address all the questions from my review. There are three small comments that I would like to see addressed, and then I would be more than happy to recommend this for publication:

1. Line 186: The authors reference a "robust standard errors method". Please include a reference for this.

2. Line 199: Revise the wording 'highest percentage of people (26.7%)' - this amount cannot be the highest percentage of people, and should be worded differently.

3. Inclusion of some of the research cited in the authors responses:

3a. In response to my comment regarding specifying whether sex differences occur by age the authors provide rich context as to the possible biological sex-related differences and this would enrich the introduction/discussion if included. Specifically, "Speaking generally, differences between males and females in youth are very common, especially in terms of cardiorespiratory fitness (doi:10.1161/CIR.0000000000000866, doi: 10.5604/01.3001.0053.7363, doi: 0.3389/fped.2021.657551). Potential explanations for this difference include sex-related differences in muscle fibre type, oxygen extraction, or the lipid content of myofibrils (doi: 10.1016/j.jsams.2008.05.006, doi: 10.1152/ajpregu.00472.2006)."

3b. In response to my comment :'Discuss why MVPA levels fell below WHO recommendations despite the

study population being a seemingly healthy cohort".

The authors state: "As a result, the absence of immediate, visible outcomes can lead to a lack of motivation to engage in physical activity during leisure time. Additionally, urban environments often lack accessible and safe spaces for outdoor activities, further limiting opportunities for movement. The widespread use of smartphones, streaming platforms, and video games has also contributed to an increasingly sedentary lifestyle among adolescents. Moreover, shifting priorities toward academic responsibilities, social interactions, and digital engagement may lead to physical activity being deprioritized in daily routines of adolescents. While these variables may hold relevance, they were not investigated as part of the present. While these variables may hold relevance, they were not investigated as part of the present study. Therefore, we could not discuss these factors in detail". I believe the findings in your article would be enriched by including the context provided in your response to my comment.

I would once again like to commend the authors for the quality of this research and for being open taking the suggested revisions. I look forward to reading this article in press!

Reviewer #2: Authors have adressed my comments. I do not have any other comment and suggest acceptance of the manuscript

**Do you want your identity to be public for this peer review?** For information about this choice, including consent withdrawal, please see our Privacy Policy

Reviewer #1: **Yes:** Elizabeth S. Dinkele

Reviewer #2: No

---

## [Author Response · Author response to Decision Letter 2]

11 Feb 2026

REVIEWER 1

I would like to commend the authors of this manuscript, and on their efforts to answer and address all the questions from my review. There are three small comments that I would like to see addressed, and then I would be more than happy to recommend this for publication:

Authors’ reply: We sincerely thank Reviewer 1 for the kind words and for taking the time to provide thoughtful feedback on our manuscript. We have carefully considered your three additional comments and have revised the manuscript accordingly.

1. Line 186: The authors reference a "robust standard errors method". Please include a reference for this.

Authors’ reply: We thank the reviewer for this comment. We have now included a reference for the robust standard errors method (White, 1980), and the Methods section has been updated accordingly. We hereby note that the reference has been added to the manuscript as reference [32]; doi.org/10.2307/1912934.

2. Line 199: Revise the wording 'highest percentage of people (26.7%)' - this amount cannot be the highest percentage of people, and should be worded differently.

Authors’ reply: We thank the reviewer for this suggestion. The wording has been revised accordingly, and the manuscript now states as follows: “The most frequently reported level of physical activity was 2–3 hours per week (26.7%).”

3. Inclusion of some of the research cited in the authors responses:

3a. In response to my comment regarding specifying whether sex differences occur by age the authors provide rich context as to the possible biological sex-related differences and this would enrich the introduction/discussion if included. Specifically, "Speaking generally, differences between males and females in youth are very common, especially in terms of cardiorespiratory fitness (doi:10.1161/CIR.0000000000000866, doi: 10.5604/01.3001.0053.7363, doi: 0.3389/fped.2021.657551). Potential explanations for this difference include sex-related differences in muscle fibre type, oxygen extraction, or the lipid content of myofibrils (doi: 10.1016/j.jsams.2008.05.006, doi: 10.1152/ajpregu.00472.2006)."

Authors’ reply: We thank the reviewer for this valuable suggestion. The context regarding possible biological sex-related differences has now been incorporated into the Discussion section, as recommended.

We hereby note that the following references have been added to the manuscript:

[4]; doi.org/10.1161/CIR.0000000000000866

[38]; doi:10.5604/01.3001.0053.7363

[39]; doi:10.14814/phy2.70616

[40]; doi:10.1016/j.jsams.2008.05.006

[41]; doi:10.1152/ajpregu.00472.2006

3b. In response to my comment :'Discuss why MVPA levels fell below WHO recommendations despite the study population being a seemingly healthy cohort".

The authors state: "As a result, the absence of immediate, visible outcomes can lead to a lack of motivation to engage in physical activity during leisure time. Additionally, urban environments often lack accessible and safe spaces for outdoor activities, further limiting opportunities for movement. The widespread use of smartphones, streaming platforms, and video games has also contributed to an increasingly sedentary lifestyle among adolescents. Moreover, shifting priorities toward academic responsibilities, social interactions, and digital engagement may lead to physical activity being deprioritized in daily routines of adolescents. While these variables may hold relevance, they were not investigated as part of the present. While these variables may hold relevance, they were not investigated as part of the present study. Therefore, we could not discuss these factors in detail". I believe the findings in your article would be enriched by including the context provided in your response to my comment.

Authors’ reply: We thank the reviewer for this constructive suggestion. The contextual discussion regarding potential motivational, environmental, and lifestyle factors influencing physical activity has now been incorporated into the Discussion section of the manuscript. These factors are presented as plausible explanations and are clearly framed as not having been directly assessed in the present study.

We hereby note that another reference has been added to the manuscript as reference [42]; doi:10.3389/fphys.2023.1131195.

REVIEWER 2

Authors have adressed my comments. I do not have any other comment and suggest acceptance of the manuscript.

Authors’ reply: We sincerely thank Reviewer 2 for the careful evaluation of our manuscript and for the positive feedback. We greatly appreciate the time and effort devoted to reviewing our work and are grateful for the recommendation for acceptance.

---

## [Editor Report · Decision Letter 2]

16 Feb 2026

The role of physical activity in modulating six-minute walk distance in adolescents

PONE-D-25-18111R2

Dear Dr. Carraro,

We’re pleased to inform you that your manuscript has been judged scientifically suitable for publication and will be formally accepted for publication once it meets all outstanding technical requirements.

Kind regards,

Andrea Martinuzzi

Academic Editor

PLOS One
---

## [Editor Report · Acceptance letter]

PONE-D-25-18111R2

PLOS One

Dear Dr. Carraro,

I'm pleased to inform you that your manuscript has been deemed suitable for publication in PLOS One. Congratulations! Your manuscript is now being handed over to our production team.

Kind regards,

on behalf of

Dr. Andrea Martinuzzi

Academic Editor

PLOS One